# Surface protein profiling of prostate-derived extracellular vesicles by mass spectrometry and proximity assays

Ehsan Manouchehri Doulabi [1,7], Claudia Fredolini[1,7], Radiosa Gallini [1], Liza Löf[1], Qiujin Shen[1], Ryoyo Ikebuchi[1,2], Louise Dubois[3], Alireza Azimi[1], Olivier Loudig[4], Susanne Gabrielsson [5], Ulf Landegren [1], Anders Larsson [3], Jonas Bergquist [6] & Masood Kamali-Moghaddam [1✉]

Extracellular vesicles (EVs) are mediators of intercellular communication and a promising class of biomarkers. Surface proteins of EVs play decisive roles in establishing a connection with recipient cells, and they are putative targets for diagnostic assays. Analysis of the surface proteins can thus both illuminate the biological functions of EVs and help identify potential biomarkers. We developed a strategy combining high-resolution mass spectrometry (HRMS) and proximity ligation assays (PLA) to first identify and then validate surface proteins discovered on EVs. We applied our workflow to investigate surface proteins of small EVs found in seminal fluid (SF-sEV). We identified 1,014 surface proteins and verified the presence of a subset of these on the surface of SF-sEVs. Our work demonstrates a general strategy for deep analysis of EVs' surface proteins across patients and pathological conditions, proceeding from unbiased screening by HRMS to ultra-sensitive targeted analyses via PLA.

[1] Department of Immunology, Genetics & Pathology, Science for Life Laboratory, Uppsala University, Uppsala, Sweden. [2] JSPS Overseas Research Fellow, Japan Society for the Promotion of Science, Tokyo, Japan. [3] Department of Medical Sciences, Clinical Chemistry, Uppsala University, Uppsala, Sweden. [4] Center for Discovery and Innovation, Hackensack Meridian Health, Nutley, NJ, USA. [5] Division of Immunology and Allergy, Department of Medicine, Karolinska Institutet, Solna, Sweden. [6] Department of Chemistry-BMC, Analytical Chemistry, Uppsala University, Uppsala, Sweden. [7] These authors contributed equally: Ehsan Manouchehri Doulabi, Claudia Fredolini. ✉email: masood.kamali@igp.uu.se

Extracellular vesicles (EVs) are lipid bilayered nanoparticles secreted by most cells. There are three main subgroups of EVs, which are classified according to their sizes, biogenesis, and density: (i) exosomes, (ii) microvesicles, and (iii) apoptotic bodies. Exosomes are small-EVs (sEVs) with a size ranging between 30 and 150 nm, which are generated by the inward budding of endosomes, which lead to the production of multivesicular bodies (MVBs). MVBs ultimately fuse with the plasma membrane and release their sEV content into the extracellular matrix as well as in bodily fluids, where sEVs have been shown to play critical roles in intercellular communication[1–5]. Microvesicles with a size range of 100–800 nm and apoptotic bodies with a size range of 200 nm–5 μm are shed directly from the plasma membranes of viable cells and those undergoing programmed cell death, respectively[6]. sEVs, as well as microvesicles, and apoptotic bodies can mediate intercellular transport for the delivery of molecular cargos containing proteins, lipids, small-RNAs and other RNA species, and genomic DNA fragments[1,7,8].

Recent studies have demonstrated that the content of EVs differs depending on their cellular lineage and that they thereby reflect the cells they originate from. Analysis of the dynamic variation of sEV fingerprints may provide a valuable means to track and monitor diseases[9–13]. Current molecular studies and assays focused on circulating biomarkers mostly evaluate the RNA and lipid contents of circulating sEVs, but there is an increasing interest also to investigate the protein composition of sEVs[3]. In particular, the surface proteins of EVs are of great interest due to their role in establishing contact with target cells, which can lead to their cellular uptake or fusion with plasma membrane before release of their molecular cargos[14].

Seminal fluid sEVs (SF-sEV), also known as prostasomes, are secreted by prostate gland into the seminal fluid, where one of their key functions is to directly interact and protect sperm cells[15–17]. Fusion of SF-sEVs with the sperm plasma membrane is required for the regulation of different aspects of sperm cell function, such as motility and capacitation, one of the last steps in the maturation of spermatozoa required to acquire fertilizing capacity[18,19]. SF-sEVs have also been implicated in the interaction between prostatic cancer cells and their microenvironment[20]. They are recognized as potential biomarkers in male infertility[21] and prostate cancer[22,23], yet little is known about the cellular mechanisms leading their production and the molecular pathways driving SF-sEV functions.

Mass spectrometry (MS)-based protein analysis is an efficient and widely used tool to characterize EVs proteins[24,25]. Data generated by MS have contributed to the development of online databases, such as ExoCarta (www.exocarta.org)[26] and Vesiclepedia (www.microvesicles.org), which list proteins found in EVs, including sEVs[27]. Several biochemical techniques have been applied for MS-based analysis of membrane proteins with low abundance in EVs[28]. In particular, non-membrane permeable reagents for chemical derivatization, such as sulfo-NHS-SS-biotin, have been implemented to study both cell surface proteins[29] and EV surface proteins from pancreatic cancer cells[30] and the HMC-1 mast cell line[31].

However, MS-based strategies alone fail to establish the correct localization and orientation of proteins on EVs' surfaces. Therefore, further validation experiments applying orthogonal methods[32,33] are often required, such as immune-affinity methods coupled to electron- and super-resolution microscopy[34–37]. Generally, enzyme-linked immunosorbent assay (ELISA) and other affinity-based assays are used to detect known surface molecules on intact EVs. Nonetheless, these methods are not suited for broad and multiplex investigations of surface proteins since only two target proteins can be interrogated per assay, and results can be compromised by non-specific binding and cross-reactivity. Although antibody arrays have recently been applied in different formats for the detection of large sets of EV-associated proteins[38–40], due to the limited spectrum of antigen recognition, they are not adapted for unbiased discovery of biologically and clinically relevant EV proteins, despite their multiplexing capacity and low sample requirements.

In this study, we developed a workflow for the discovery and validation of unknown EV surface proteins by combining high-resolution MS (HRMS) analysis of biotin-derivatized surface proteins with flow cytometry-based proximity ligation assays (Exo-PLA) and/or solid-phase proximity ligation assays (SP-PLA) (Fig. 1). Flow cytometry is a robust technique to measure surface cell markers and a tool used routinely for cell profiling in clinical practice[41,42]. This technique also lends itself to analyses of arrays of beads[43,44]. The possibility of distinguishing EVs' subpopulations makes flow cytometry particularly attractive for the investigation of sEVs; however, due to their small size and the low number of surface molecules, opportunities for using conventional flow cytometry remain limited. The use of Exo-PLA, a multicolor detection, and signal amplification technology, makes it possible to overcome the size and signal limitations for flow cytometric analysis of sEVs. Indeed, taking advantage of local signal amplification via rolling circle amplification (RCA), individual sEVs become detectable well above the flow cytometric cut-off. By using different affinity binders, such as antibodies, and several fluorescent dyes, different subpopulations of sEVs can then be visualized and enumerated[45,46]. SP-PLA relies on the recognition of the target by combined sets of three antibodies, with readouts via real-time PCR, which provides excellent sensitivity and specificity for the detection of sEVs in solution[23].

We applied this workflow to investigate SF-sEVs as essential determinants of male fertility and potential cancer biomarkers. In this study, we sought to expand our current knowledge on surface proteins from SF-sEVs, thereby allowing us to evaluate their biological role and to identify and classify SF-sEVs on the molecular level for diagnostic purposes.

## Results

**Purification and characterization of seminal fluid and PC3-derived sEVs.** SEVs from human seminal fluid (SF-EVs) and from the prostate cancer cell line PC3 were purified according to optimized protocols matching each matrix[47]. The procedure involved a combination of ultracentrifugation, size exclusion chromatography, and sucrose gradient separation. The quality of the purified vesicles was examined by negative stain transmission electron microscopy (TEM), western blot, and nanoparticle tracking analysis (NTA) as recommended by Minimal Information for Studies of Extracellular Vesicles (MISEV) 2018 guidelines[48]. Negative stain EM revealed structurally intact SF-sEVs and PC3 sEVs (Fig. 2a). Western blot analysis was performed to demonstrate the presence of sEV markers CD9, CD63, CD81 and tumor susceptibility gene 101 protein (TSG-101), and the absence of the endoplasmic reticulum (ER) marker calnexin, indicating the purity of the sEV samples from protein contaminations (Fig. 2b). NTA revealed an average particle diameter of 190 and 160 nm for SF-sEVs (i.e., from seminal fluid) and PC3 sEVs, respectively, with a mean concentration of $1.3 \times 10^9$ particles/ml and $1.4 \times 10^9$ particles/ml, respectively (Fig. 2c).

**Purification and surface protein profiling of seminal fluid and PC3-derived sEVs.** The surface proteins of intact SF-sEVs and PC3 sEVs were labeled using sulfo-NHS-SS-biotin, a membrane-impermeable reagent reacting with primary amines[30]. The fractions obtained from the purification process were: (i) protein in total sEV lysate (Total); (ii) proteins isolated from sEV surfaces (Surface), and

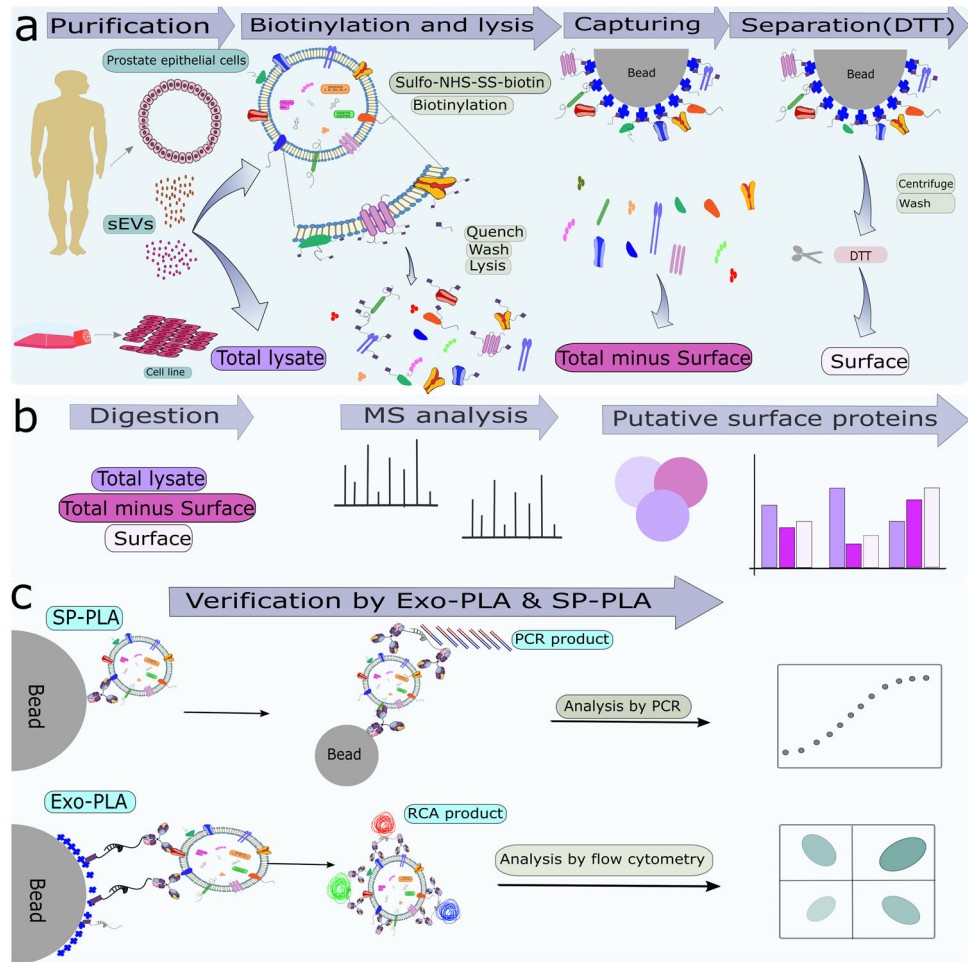

**Fig. 1 Schematic illustration of the strategy to identify and validate sEV surface proteins. a** SEVs were isolated from human seminal fluids and PC3 culture media, respectively, and were treated with sulfo-NHS-SS-biotin to biotinylate outer membrane proteins on the surface of the sEVs before addition of lysis buffer. Biotinylated surface proteins were captured on streptavidin beads, and were released by DTT. **b** Total, total minus surface, and surface proteins were digested and identified by label-free semi-quantitative HRMS analysis. **c** The presence of these proteins was validated by Exo-PLA and SP-PLA.

(iii) supernatant proteins left after isolation of surface proteins (Total minus Surface) (Figs. 1a and 2d). Proteins from each fraction were digested by trypsin and analyzed by HRMS (Fig. 1b). A total PC3 cell lysate was prepared and analyzed as a control. A complete list of proteins identified across all the fractions for SF-sEV, PC3 sEVs and PC3 cell lysate is reported in Supplementary Data 1. For all the proteins identified in this study, the table in Supplementary Data 1 reports: 1- Gene Ontology (GO) annotation; 2- localization; 3- involvement in biological processes, and 4- molecular function, and we provide an annotation regarding tissue specificity and pathways using data downloaded from the UniProt Knowledgebase database (UniProtKB). Isolated surface proteins identified by HRMS were confirmed to be expressed on the sEV surfaces using Exo-PLA and SP-PLA (Fig. 1c). The quality of sEV purification was evaluated by gel electrophoresis (Fig. 2d). Further, in order to assess the robustness of our workflow, we prepared and analyzed the SF-sEV samples in replicates: (see "Methods", Supplementary Fig. 1a–c and Supplementary Fig. 2b–g). The protocol for sample preparation demonstrated low variability between the replicates (7–19%). When comparing technical replicates for the same biological samples (Rep 2 and Rep 3), 1086 out of a total 1364 proteins were found to be in common for total minus surface (Supplementary Fig. 1b) and 653 out of a total 915 for surface (Supplementary Fig. 1c). When technical and biological replicates

were compared for total minus surface, 41 proteins were found in common for Rep 1 and Rep 2, and 64 in common between Rep 1 and Rep 3 (Supplementary Fig. 1b). For surface Rep 1 vs. Rep 2, 46 common proteins were identified, while the number of shared proteins for Rep 1 vs. Rep 3 was 31 (Supplementary Fig. 1c). A high correlation between relative protein abundance in normalized peptide spectral matches (nPSMs) was found between replicates (Pearson's r: 0.84–0.99; Supplementary Fig. 2).

After merging proteins identified in the replicates, the analysis for the fractions total lysate, surface and total minus surface resulted in the identification of 1414, 1014, and 1460 proteins, respectively (Supplementary Fig. 1, Supplementary Data 1, and Fig. 3b). As shown in Fig. 3, 875 proteins were identified across all samples, 381 proteins were in common between total lysate and total minus surface, and 20 and 39 proteins were in common between surface and total minus surface and total lysate, respectively.

**Characterization of the identified proteins.** In order to perform a semi-quantitative proteomics analysis across all seminal fluid and PC3 sEVs fractions, the number of peptide spectral matches (PSMs) associated with each identified protein was normalized according to the total number of PSMs identified in

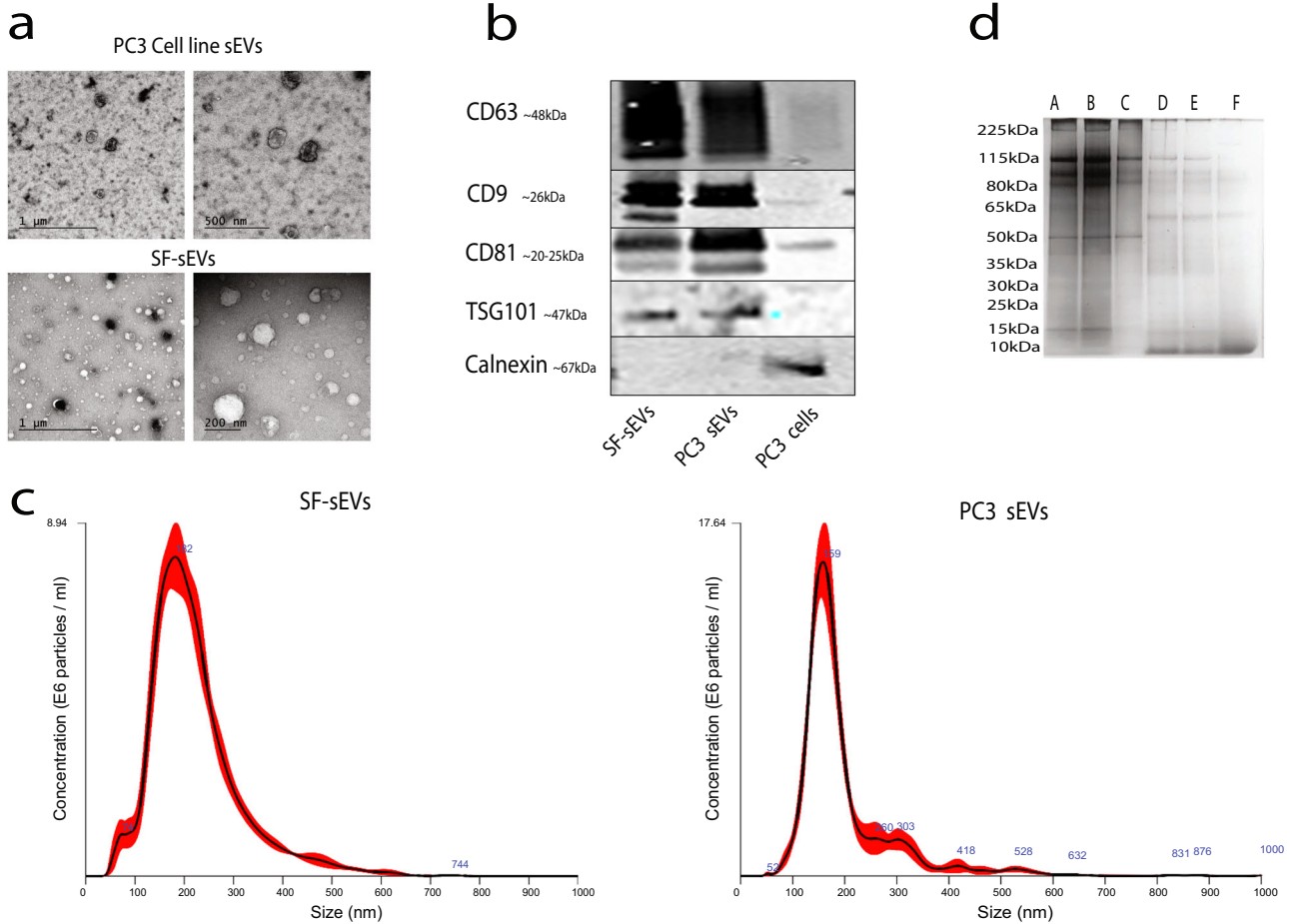

**Fig. 2 Quality control of seminal fluid and PC3 sEVs. a** Negative stained-EM of SF-sEVs and PC3 sEVs. **b** Western blot results for sEV markers TSG-101, CD63, CD81, and CD9. The ER marker calnexin was targeted as a negative control. **c** NTA analysis of the mode particle diameter recorded values of 155.8 and 192.2 nm for SF-sEVs and PC3 sEVs, respectively. **d** The quality of the purification was evaluated by gel electrophoresis for PC3 sEVs and SF-sEVs Repl 2 and Repl 3. A: Total minus surface fraction PC3 sEVs; B: Total minus surface fraction SF-sEVs Repl 2; C: Total minus surface SF-sEVs fraction Repl 3; D: Surface SF-sEVs fraction Repl 2; E: Surface SF-sEVs fraction Repl 3; F: Surface fraction PC3 sEVs.

each sample to obtain the nPSM. Among the fifty most abundant proteins identified in the surface fraction of SF-sEVs were semenogelin-1 (SEMG1), semenogelin-2 (SEMG2), fibronectin (FN1), CD13, CD10, fatty acid synthase (FASN) and creatine kinase B (CKB). SEMG1 and SEMG2 were particularly abundant with nPSM values of 8.1 and 5, respectively (Table 1 and Supplementary Data 1). Our analysis resulted in identification of 273 new putative sEVs proteins, currently not listed as sEV proteins in the Exocarta database nor in the UniProtKB (Fig. 3a). Identification of surface proteins by means of protein labeling using sulfo-NHS-SS-biotin on intact sEVs does not assure a 100% efficiency in surface protein extraction, and some degree of contamination between the fractions could be expected. Therefore, in order to identify the most highly enriched proteins in each fraction, fold change ranking and t-test statistics were used. Proteins found uniquely or more abundantly in the surface fraction compared to those in total minus surface fraction (fold change (FC) >2 and/or $P$-value <0.05) were defined as surface enriched. Proteins enriched in the total minus surface fraction compared to those in the surface fraction were defined as cargo enriched. The analysis resulted in the identification of 74 surface enriched SF-sEV proteins of which 43 were classified as membrane proteins according to the GO cellular component classification (Supplementary Data 1 and 2).

The functions and network of 74 surface enriched proteins were analyzed using STRING, a search tool and biological database for the detection of genes/proteins and for the prediction of protein-protein interactions (https://string-db.org/). The proteins were analyzed based on seven criteria: (1) expression in cytosol, (2) putative involvement in disease development, (3) roles in immune system, (4) any putative function in male reproductive system, (5) presence of proteins in EVs, (6) seminal vesicles and (7) multivesicular body. Brief functional descriptions for these proteins are listed in Supplementary Data 3 and the fulfilling of one or more criteria for each protein is illustrated in Supplementary Fig. 3.

The most enriched proteins in the surface fraction compared to total minus surface were KRT2, DNAJC3, and CASP14 (Fig. 4a, b, Supplementary Data 2; three replicates). Known sEV markers among the more abundant proteins in the surface fraction of SF-sEVs were ANPEP, FASN, and CLU (Fig. 4c).

Furthermore, we analyzed subcellular localization, molecular function, and biological process GO terms for the proteins that we found to be either surface or cargo-enriched (Supplementary Data 1). For both SF-sEVs and PC3 sEVs, subcellular localization analysis showed strong enrichment in proteins annotated as being extracellular among proteins that were more abundant in the surface fraction. Proteins associated with the extracellular region were 3% of the cargo fraction, while up to 13% of proteins in the

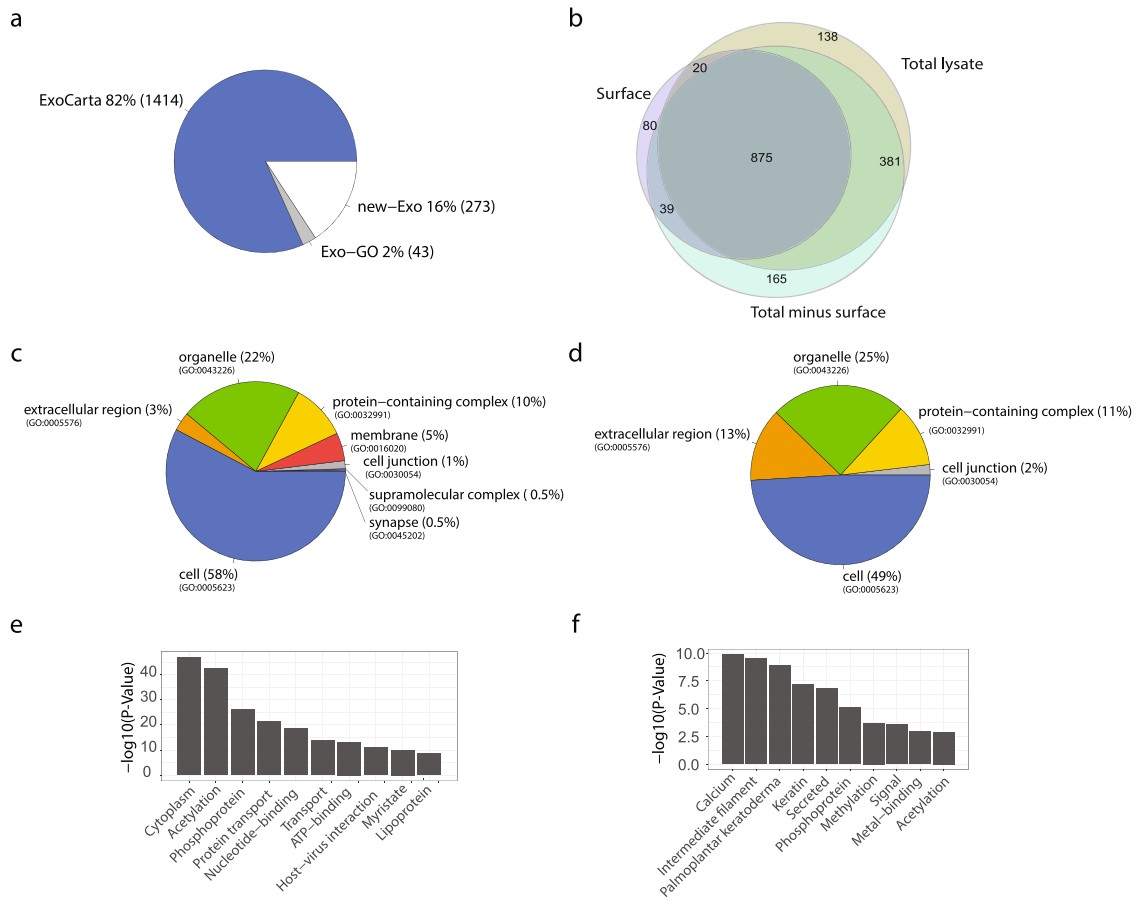

**Fig. 3 Global proteomic analysis and GO annotation of SF-sEVs proteins. a** Proteins identified by MS analysis of SF-sEVs and PC3 sEVs were compared against the ExoCarta and GO databases. **b** Venn diagram representation of the proteins identified in the three SF-sEVs fractions. **c** Pie chart representation of Gene Ontology Cellular Component categories (GO-CC) for proteins enriched in total minus surface. **d** GO-CC categories for proteins enriched in surface fractions. **e** Top ten terms obtained by functional annotation analysis in DAVID for proteins enriched in total minus surface fractions. **f** Top ten terms identified in DAVID for proteins enriched in surface fractions. Only proteins identified in at least two samples were included in the analysis.

surface fraction were annotated as surface proteins (Fig. 3c, d and Supplementary Figs. 4 and 5). Similarly, the analysis of GO terms for molecular functions and biological processes revealed distinct specific traits for the surface and the cargo proteins (Supplementary Fig. 4), which are summarized by the functional annotation analysis performed using the DAVID database (Fig. 3e, f). The functional GO analysis revealed that the terms highly enriched for the surface fraction were calcium-binding and intermediate filament, while cytoplasm and acetylation were top terms for the cargo-enriched proteins. The top ten biological terms obtained by DAVID for each GO category are listed in Supplementary Fig. 4.

**Verification of identified surface proteins by Exo-PLA and SP-PLA.** The presence of the proteins identified by MS on the surface of intact SF-sEVs and PC3 sEVs was further investigated by developing specific Exo-PLA or SP-PLA tests. Exo-PLA visualizes sEVs using multiple pairs of antibody probes. When both members of a pair of probes bind closely to one another on the surface of the sEVs, a circular DNA strand can be physically generated, which gives rise to an RCA product. The repeated sequences of the RCA product are then visualized using hybridization probes, labeled with fluorescent dyes, and detected by flow cytometry. This powerful tool enables molecular differentiation of sEVs subpopulations solely based on the combination of proteins expressed on their surface[45]. The gating strategy is explained in Supplementary Fig. 6a. We used a combination of Exo-PLA probes targeting known and highly expressed sEV markers as controls and selected targets identified through our HRMS analysis to confirm the presence of SEMG1, prostatic acid phosphatase (ACPP), prostate-specific antigen (PSA), prostate-specific membrane antigen (PSMA), prostaglandin D2 (PTGDS), CD59, A-kinase anchor protein (AKAP4), cysteine-rich secretory protein 1 (CRISP1), and glyceraldehyde-3-phosphate dehydrogenase testis-specific (GAPDS) on the surface of SF-sEVs and PC3 sEVs (Fig. 5a). In particular, we identified two distinct subpopulations of SF-sEVs expressing SEMG1 either with CD59 or with PSMA, while for PC3 sEVs, expression of SEMG1 and CD59 was less abundant. A combination of Exo-PLA probes directed against SEMG1, PSMA, and PTGDS revealed double- and triple-positive populations for both SF-sEVs and PC3 sEVs. A combination of antibody conjugates directed against SEMG1, GAPDS, and AKAP4 identified double- and triple-positive populations on SF-sEVs but not on PC3 sEVs, (Fig. 5a), confirming the MS data, where the highest abundance of AKAP4 was found in the fraction with the surface proteins of SF-sEV compared to that from PC3 sEVs. Double-positive populations of SEMG1 and ACPP and SEMG1 and CRISP1 were also found in both SF-sEVs and PC3 sEVs; however, triple-positive populations in SF-sEVs appear to be much less abundant compared to the population positive for SEMG1 together with PSMA and PTGDS or SEMG1 with AKAP4 and GAPDS. As an additional control for the different fluorescent signals, the samples were also examined

**Table 1 Fifty-three proteins found more abundant in the surface fraction from SF-sEVs.**

| Accession | Gene name | Description | # Repl. | PSM (Av) | nPSM (Av) |
|---|---|---|---|---|---|
| P04279 | SEMG1 | Semenogelin-1 | 3 | 628.3 | 8.10 |
| Q02383 | SEMG2 | Semenogelin-2 | 3 | 394.0 | 5.05 |
| P15144 | ANPEP | Aminopeptidase N | 3 | 133.7 | 1.74 |
| P02751 | FN1 | Fibronectin | 3 | 102.7 | 1.35 |
| P08473 | MME | Neprilysin (CD 10) | 3 | 83.3 | 1.09 |
| P49327 | FASN | Fatty acid synthase | 3 | 77.7 | 0.98 |
| P12277 | CKB | Creatine kinase B-type | 3 | 72.7 | 0.95 |
| P10909 | CLU | Clusterin | 3 | 71.7 | 0.93 |
| P04264 | KRT1 | Keratin, type II cytoskeletal 1 | 3 | 71.7 | 0.90 |
| P60709 | ACTB | Actin, cytoplasmic 1 | 3 | 65.7 | 0.86 |
| P49221 | TGM4 | Protein-glutamine gamma-glutamyltransferase 4 | 3 | 66.3 | 0.85 |
| P02768 | ALB | Serum albumin | 3 | 65.7 | 0.84 |
| P15309 | ACPP | Prostatic acid phosphatase | 3 | 64.0 | 0.83 |
| P27487 | DPP4 | Dipeptidyl peptidase 4 | 3 | 63.0 | 0.82 |
| P13645 | KRT10 | Keratin, type I cytoskeletal 10 | 3 | 59.7 | 0.75 |
| P12273 | PIP | Prolactin-inducible protein | 3 | 57.0 | 0.73 |
| P07900 | HSP90AA1 | Heat shock protein HSP 90-alpha | 3 | 55.3 | 0.71 |
| P08238 | HSP90AB1 | Heat shock protein HSP 90-beta | 3 | 52.3 | 0.67 |
| P02788 | LTF | Lactotransferrin | 3 | 50.7 | 0.64 |
| O43451 | MGAM | Maltase-glucoamylase, intestinal | 3 | 49.0 | 0.63 |
| P11142 | HSPA8 | Heat shock cognate 71 kDa protein | 3 | 45.3 | 0.58 |
| P0DMV9 | HSPA1B | Heat shock 70 kDa protein 1B | 3 | 43.7 | 0.56 |
| P35908 | KRT2 | Keratin, type II cytoskeletal 2 epidermal | 3 | 41.7 | 0.53 |
| P35579 | MYH9 | Myosin-9 | 3 | 41.0 | 0.53 |
| Q8WUM4 | PDCD6IP | Programmed cell death 6-interacting protein | 3 | 41.0 | 0.53 |
| P35527 | KRT9 | Keratin, type I cytoskeletal 9 | 3 | 41.3 | 0.52 |
| P80723 | BASP1 | Brain acid soluble protein 1 | 3 | 39.7 | 0.51 |
| P13796 | LCP1 | Plastin-2 | 3 | 39.3 | 0.51 |
| P07288 | KLK3 | Prostate-specific antigen | 3 | 36.3 | 0.47 |
| P0CG48 | UBC | Polyubiquitin-C | 3 | 35.0 | 0.46 |
| Q9Y3R5 | DOP1B | Protein dopey-2 | 3 | 35.0 | 0.44 |
| P62258 | YWHAE | 14-3-3 protein epsilon | 3 | 33.7 | 0.44 |
| P68371 | TUBB4B | Tubulin beta-4B chain | 3 | 33.7 | 0.43 |
| P14618 | PKM | Pyruvate kinase PKM | 3 | 32.3 | 0.41 |
| Q96KP4 | CNDP2 | Cytosolic non-specific dipeptidase | 3 | 31.7 | 0.41 |
| Q14204 | DYNC1H1 | Cytoplasmic dynein 1 heavy chain 1 | 3 | 32.7 | 0.41 |
| P04406 | GAPDH | Glyceraldehyde-3-phosphate dehydrogenase | 3 | 31.7 | 0.40 |
| O14494 | PLPP1 | Phospholipid phosphatase 1 | 3 | 30.0 | 0.40 |
| P15311 | EZR | Ezrin | 3 | 30.7 | 0.39 |
| P68104 | EEF1A1 | Elongation factor 1-alpha 1 | 3 | 30.7 | 0.39 |
| O95716 | RAB3D | Ras-related protein Rab-3D | 3 | 29.0 | 0.38 |
| O00194 | RAB27B | Ras-related protein Rab-27B | 3 | 28.7 | 0.37 |
| P04792 | HSPB1 | Heat shock protein beta-1 | 3 | 27.7 | 0.36 |
| P00738 | HP | Haptoglobin | 3 | 28.0 | 0.36 |
| O43707 | ACTN4 | Alpha-actinin-4 | 3 | 27.3 | 0.35 |
| P04075 | ALDOA | Fructose-bisphosphate aldolase A | 3 | 27.3 | 0.35 |
| P07437 | TUBB | Tubulin beta chain | 3 | 27.7 | 0.35 |
| P00558 | PGK1 | Phosphoglycerate kinase 1 | 3 | 27.3 | 0.35 |
| P02538 | KRT6A | Keratin, type II cytoskeletal 6A | 1 | 30.0 | 0.13 |
| Q9BQE3 | TUBA1C | Tubulin alpha-1C chain | 1 | 28.0 | 0.12 |
| O95359 | TACC2 | Transforming acidic coiled-coil-containing protein 2 | 3 | 27.0 | 0.36 |
| P54652 | HSPA2 | Heat shock-related 70 kDa protein 2 | 3 | 27.3 | 0.35 |
| P55072 | VCP | Transitional endoplasmic reticulum ATPase | 3 | 26.7 | 0.35 |

#Repl.: Number of replicate MS samples in which the protein was identified; PSM (Av): Averaged number of PSMs in the replicates; nPSM (Av): normalized PSMs averaged.

by fluorescence microscopy for visualization of fluorescent RCA products and final validation (Supplementary Fig. 6b).

The immunoaffinity-based SP-PLA offers the opportunity to detect and quantify intact sEVs by targeting up to three surface proteins. In this experimental setting, SF-sEVs were first captured using antibodies against one of the targeted proteins CRISP1, SEMG1, PTGDS, AKAP4, or GAPS. The captured SF-sEVs were then detected using a pair of oligonucleotide-conjugated antibody probes directed against the known

markers CD9 and CD26. Higher detection signals were recorded when SF-sEVs were captured by antibodies directed against CRISP1, SEMG1, or PTGDS antibodies compared to SF-sEV captured by antibodies against AKAP4 and GAPDS. Nonetheless, all the targets were robustly detectable on the surface of SF-sEVs (Fig. 5b). As expected, a negative control experiment, using a capture antibody targeting the ER marker calnexin, gave undetectable signals even at high sample concentrations (Fig. 5b). The SP-PLA data were in line with

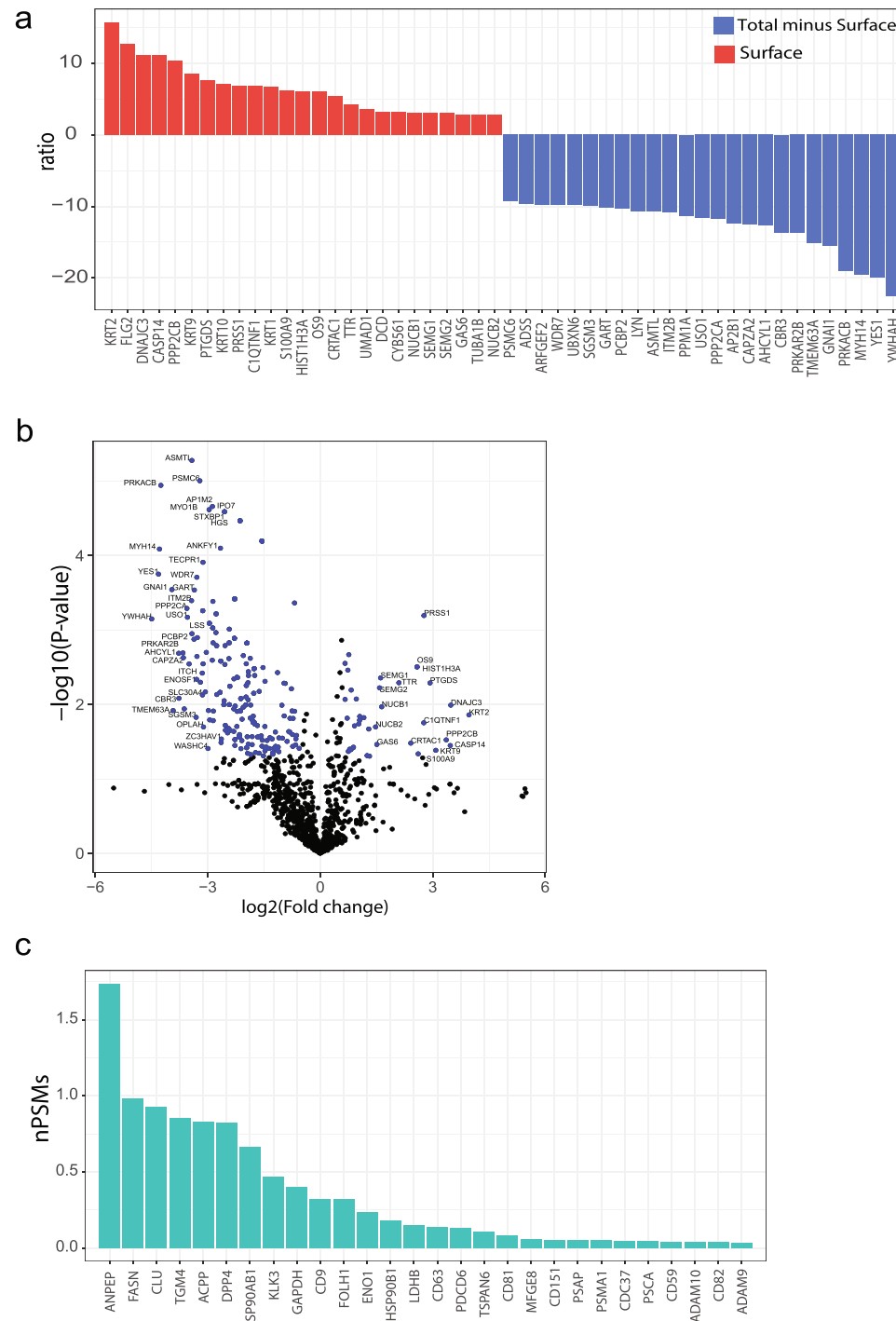

**Fig. 4 Surface proteins expression in SF-EVs. a** Proteins identified in the surface and in total minus surface fractions were compared by fold change and t-test statistical analysis. **a** The bar plot represents the ratio of the 25 most enriched proteins in surface and in total minus surface fractions. **b** The volcano plot represents fold changes and *P*-value for proteins identified at least in two samples. **c** Abundance in normalized peptide spectrum matches (nPSMs) of known sEV markers in the surface fraction of SF-sEVs.

those obtained by Exo-PLA, further supporting the surface expression of proteins identified in HRMS analyses.

**Discussion**
Information about the identity of proteins expressed on the membrane of sEVs can be of value for diagnostic and preparative procedures. In this study, we developed a state-of-the-art strategy

to investigate surface proteins of sEVs, by combining unbiased profiling using HRMS with Exo-PLA[45] and SP-PLA[49] to validate the presence of the identified proteins on the surface of intact sEVs. The use of an unbiased MS approach allowed us to identify in total 1730 proteins in SF-sEVs and PC3 sEVs, of which 273 had not previously been reported, thus providing a list of potential targets for future studies of circulating SF-sEVs and other sEVs (Fig. 3a).

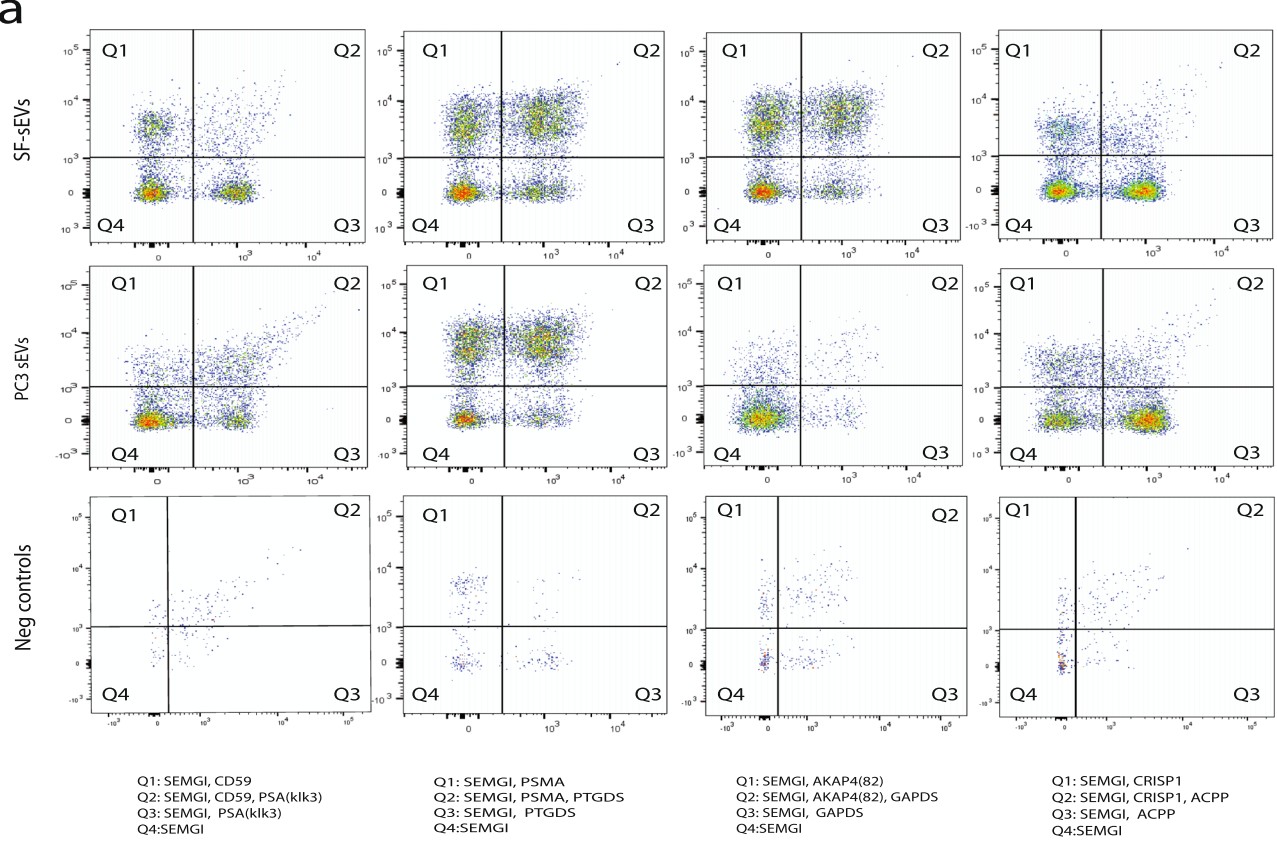

Q1: SEMGI, CD59
Q2: SEMGI, CD59, PSA(klk3)
Q3: SEMGI,  PSA(klk3)
Q4: SEMGI

Q1: SEMGI, PSMA
Q2: SEMGI, PSMA, PTGDS
Q3: SEMGI,  PTGDS
Q4: SEMGI

Q1: SEMGI, AKAP4(82)
Q2: SEMGI, AKAP4(82), GAPDS
Q3: SEMGI,  GAPDS
Q4: SEMGI

Q1: SEMGI, CRISP1
Q2: SEMGI, CRISP1, ACPP
Q3: SEMGI,  ACPP
Q4: SEMGI

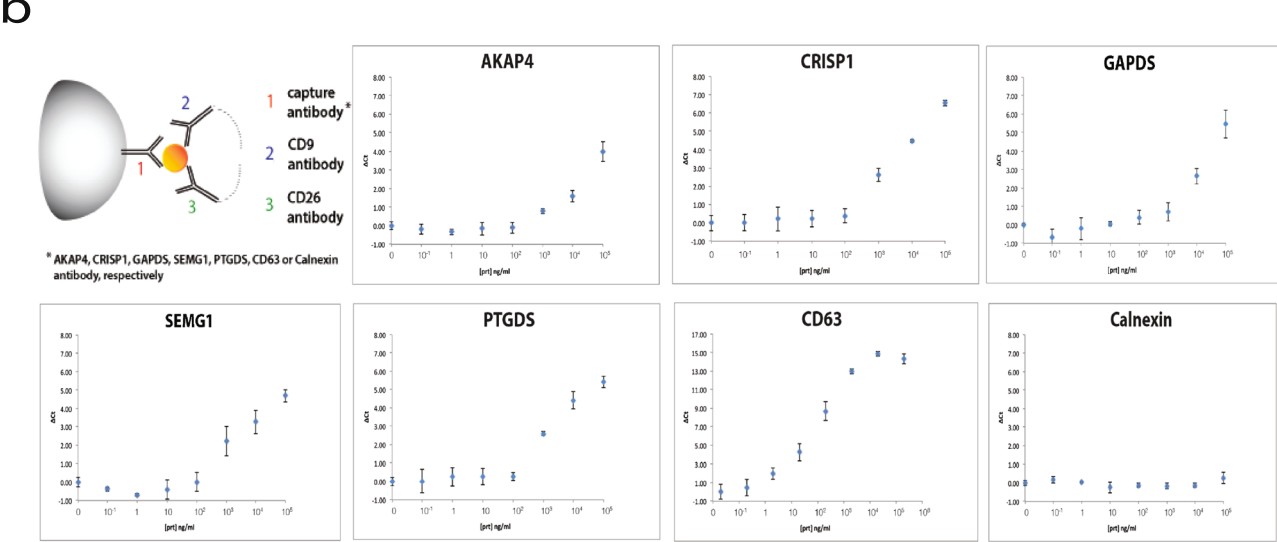

**Fig. 5 Validation of surface protein markers on SF-sEVs and PC3 sEVs by Exo-PLA and SP-PLA. a** Combination of different markers on the surface of sEVs detected by Exo-PLA. Gating of positive signals in flow cytometry and different populations are displayed. **b** SF-sEVs were captured by beads coated with antibodies for specific targets identified by HRMS analyses (CRISP1, AKAP4, GAPDS, SMG1, and PTGDS), indicated for each panel, and detected with PLA probes against CD9 and CD26 markers. The abundant marker CD68 was used as capture for positive control of assay performances, the ER marker calnexin was used as negative control capture antibody. The x-axes display the concentration of samples in ng of total protein per ml. The y-axes display the differences between the threshold cycle for the real-time PCR readout of SP-PLA reactions for the samples and the threshold cycle of the negative control (assay buffer without samples). The error bars are representing standard deviation (SD), each sample was run in triplicate.

The strategy described herein provides an efficient means to study cell surface proteins as well as EVs surface proteins[29]. This approach served to identify a total of 1014 putative surface proteins on SF-sEVs, where 457 proteins were found in all three replicate samples and 730 in at least two replicates (Supplementary Fig. 1). Out of the 1014 proteins, our data identified 74 unique proteins that were found solely enriched on the surface of sEVs when compared to total minus surface protein fractions (Supplementary Data 1 and 2). Proteins identified in more than one replicate sample prepared for each fraction represent the most robust discoveries and reasonably also the highest abundant proteins in that fraction. However, we still included the PLA-based validation steps for three proteins that were found in three replicate samples (SEMGI, PTGDS, and CRISP1), for one found in two replicates (AKAP4), and for one found in only one replicate (GAPDS).

SEVs released by a given tissue or cell-type represent a heterogeneous population of EVs[50]. In fact, seminal SF-EVs have been reported to originate from several different cell types in the male reproductive system[51,52], perhaps contributing to heterogeneous protein expression as well as heterogeneous functions. A global proteomic analysis of the data revealed that 74 surface proteins were enriched in extracellular proteins of SF-sEVs (Supplementary Data 2 and Fig. 3), which was consistent with previous observations by Webber and colleagues[53]. Using a SOMAscan array for protein measurement, the authors demonstrated that proteins considered to be secreted by the prostate might, in fact, be present on the surface of EVs. Our data, which focused on the evaluation of sEV surface proteins, therefore, reinforces the hypothesis that some of the proteins measured in biofluids may also or solely be exposed on the surface of sEVs.

Since gene expression and protein synthesis in sperm cells is interrupted before the end of spermatogenesis, sperm cells have been reported to acquire part of the molecules required for their full functions from SF-EVs, which have been shown to transfer essential proteins by fusion with the plasma membrane[54]. Spermatozoa capacitation is a process where $Ca^{2+}$ signaling is critical. As hypothesized by Park et al., sperm cells may also use other mechanisms, which do not involve ion channels for $Ca^{2+}$ signaling[55]. Such mechanisms may depend on the fusion of the sperm cell membrane with SF-EVs, which carry receptors and enzymes required for $Ca^{2+}$ mobilization. It has previously been shown that the transfer of CD38 from SF-EV into the sperm can trigger intracellular $Ca^{2+}$ release from the ryanodine receptor (53). In agreement with Park's hypothesis, our data would suggest that proteins facilitating calcium ion binding are among the most highly enriched proteins in the surface fraction of SF-sEVs (Fig. 3f and Supplementary Fig. 4). Five proteins out of the 74 found particularly enriched on the surface of SF-sEVs, CRTAC1, AGRN, GAS6, NUCB2, NUCB1, and FLG2, are annotated on GO as Calcium-binding proteins (Supplementary Data 1).

The identification of a large number of proteins in the SF-sEVs, which are also known to be expressed in the central nervous system (Supplementary Data 1) (such as kinesin heavy chain isoform 5C (KIF5C), synaptic vesicle membrane protein VAT-1, 14-3-3 protein theta (YWHAQ), breast carcinoma-amplified sequence 1, elongation factor 1-alpha 1 (EEF1A1), and brain acid-soluble protein 1 (BASP1)), or the presence of proteins highly expressed in the neuroendocrine prostatic epithelium (prostate stem cell antigen (PSCA)) supports the previously suggested neuroendocrine origin of SF-EVs and their known potential effect as neurotransmitters[56,57]. However, proteins such as the CatSper receptor[55], chromogranin B, and neuropeptide Y[56] were not identified in this study.

A subset of the surface proteins identified by HRMS was selected for validation using antibody-based methods capable of detecting intact sEVs via externally exposed surface proteins. PLA represents a unique and powerful molecular assay providing the capability for detecting protein combinations by co-localizing them in proximity using two or more probes consisting of antibodies conjugated to DNA oligonucleotides. Such probe binding in proximity can generate DNA templates for amplification by PCR or RCA and signal amplification for visual detection/confirmation. One of the PLA formats, 4PLA, has been successfully applied to demonstrate elevated levels of SF-sEVs in plasma from patients with prostate cancer[23]. Therefore, the combination of Exo-PLA and SP-PLA methods provides complementary information and the unique opportunity for the identification of surface proteins that may be targeted by diagnostic assays. While the SP-PLA allows for the identification of sEVs purified from sEV subpopulations by identification/validation of the protein surface composition, the Exo-PLA assay provides sensitivity for the quantification of sEVs from biofluids.

Using Exo-PLA, we confirmed the presence of SEMG1, ACPP, PSA, PSMA, PTGDS, AKAP4, CRISP1 and GAPDS on the surface of SF-sEVs and PC3 sEVs. PSA, ACPP, and PSMA are well known for their role in prostate physiology and as prostate cancer biomarkers. The presence of PSA on the surface of SF-EVs has been reported previously[58], whereas ACPP and PSMA, identified in this study, are integral membrane proteins[59], and PSMA has only been identified in the membrane of lysosomes[60]. Less is known about SEMG1, AKAP4, CRISP1, GAPDS, and PTGDS, which have not, to the best of our knowledge, been previously reported to be localized on the surface of SF-sEVs or any other sEVs.

SEMG1 and 2 are known to be major components of human semen coagulum and of seminal plasma (20–40% of protein content)[61,62]. Protein expression analyses have revealed that SEMG1 is mostly expressed in the glandular epithelium of seminal vesicles and some studies have demonstrated their expression in prostate cancer cells[63,64]. Yang and colleagues have already demonstrated that these proteins are integral constituents of SF-EVs[25]. SEMG 1 has been found in complex with both the glycosylphosphatidylinositol (GPI)-anchored and soluble CD52, and it has been hypothesized that GPI-anchored CD52 serves as a dock for SEMG1 in anchoring the sperm cells to form clots[65], later degraded by PSA to enable sperm motility. Recent studies found a negative correlation between sperm motility and the proportion of SEMGs bound spermatozoa[66,67]. These findings suggest that unbound sperm may be a relevant parameter for in vivo fertilization[68].

Although GAPDS, AKAP4 and PTGDS have not been previously known to be expressed on the surface of SF-EVs, they have all been identified as sperm cell surface-associated proteins[69,70], both involved in spermatozoa motility[71,72] and egg cell fertilization of mammalians[73]. It is therefore, plausible that SEMG1 and other proteins identified in our study may indeed be transferred from prostatic and epidymal cells to the surface of sperm cells in order to contribute to enable and/or enhance their physiological function.

The approach to isolate surface proteins after biotin labeling may have some limitations. First, the efficiency of biotin labeling, a process that relies on the availability of primary amines, may vary among proteins. Second, a purification using streptavidin-coated beads may be partial or incomplete, but with uniform representation, with a risk of non-specific binding to the beads[74–76]. This could, for instance, be the reason why the general sEV/SF-sEV markers, such as PSMA1, CD9, and CD151, were detected in all the analyzed fractions (Surface, Total minus Surface, and Total lysate), but were enriched in the Surface fractions (Fig. 4 and Supplementary Data 1). Third, the identified surface proteins may include some absorbing layer proteins, known as protein corona, present in body fluids or culturing media, which bind to the surface of EVs, where they can perform a specific

function or become a source of protein contaminations when studying membrane proteins[77,78].

By developing this assay, we demonstrate the feasibility of measuring surface proteins and particularly those of SF-sEVs, which may provide potential targets for the development of new diagnostic assays and the detection of prostate cancer[23]. Our analyses may provide the basis for the development of diagnostic tools for the evaluation of male fertility by providing a molecular screen that investigates the mechanisms required for sperm cells to become fully functional[79]. Our investigation of SF-sEVs surface proteins described here, presents a proof of concept for a workflow, combining the power of unbiased MS proteomic analysis with targeted PLAs for the identification of SF-sEVs proteins but with potential application to any type of EVs. SP-PLA offers the unique possibility to quantify EVs with high-sensitivity and specificity, while Exo-PLA, featuring individual EV- and multi-parametric analysis, has the potential to become an optimal platform to support a deep investigation of the role of putative surface proteins across patients and pathological conditions. Both of these assays have potential to be directly transferable for clinical applications.

## Methods

**Purification of SF-sEVs**. Seminal plasma was collected at the Reproductive Center at Uppsala University Hospital according to existing routines and under Internal Review Board authorization[8]. The two anonymized samples of SF-sEVs (samples 1 and 2) analyzed in this study were each obtained by pooling seminal plasma samples from 5 individuals. The sample collection was approved by the Ethics Committee of Uppsala University (Ups 01-367) and informed consent was obtained. The SF-sEVs were purified using an optimized protocol for isolation of sEVs from seminal fluids[80]. Human seminal plasma was centrifuged at $3000 \times g$ for 10 min and then $10,000 \times g$ for 30 min at 4 °C to pellet debris. This was followed by ultracentrifugation of the supernatant at $100,000 \times g$ for 2 h at 4 °C. The EV-containing pellet was resuspended in phosphate-buffered saline (PBS) and further purified by size-exclusion chromatography on a Superdex 200 gel-filled XK16/70 column (GE Healthcare). This was followed by a density gradient separation, where the SF-sEVs were recovered in the density range 1.13–1.19 g/ml. The concentration of the purified SF-sEVs was adjusted to 2 mg/ml using Pierce BCA protein assay (ThermoFischer Scientific) and kept at −80 °C until use.

**Cell culture and sEV purification**. The human prostate cancer cell line PC3 (ATCC-CRL1435) was cultured in RPMI 1640 medium, supplemented with 2 mM L-Glu, 100 U/ml penicillin, 100 μg/ml streptomycin, and 10% fetal bovine serum (FBS; Gibco, ThermoFisher Scientific, USA). Cells were cultured to 70–80% confluence, the media was then replaced with media containing 10% EV-depleted FBS (System Bioscience, Palo Alto, CA, USA), and the cells were grown for an additional 48 h. The cells were removed and conditioned media were collected by centrifuging at $300 \times g$ for 10 min. The conditioned media were passed through 0.22 μm filters (Merck Millipore, Burlington, Massachusetts, USA) to remove cell debris, followed by ultra-centrifugation (Beckman Coulter) at $112,000 \times g$ for 120 min in a Type SW-28 rotor to pellet sEVs. The supernatant was carefully removed, and the pellet was resuspended in 1 ml ice-cold 1× PBS, supplemented with 1× protease inhibitor (Complete Mini®, Roche, Basel, Switzerland). The sEVs were then loaded on a chromatography column and separated using the same procedure described for SF-sEVs[45]. Fractions containing sEVs were pooled and ultra-centrifuged twice at $112,000 \times g$ for 120 min using SW-28 type rotor. The resulting sEV pellet was resuspended in 200 μl of PBS supplemented with protease inhibitors and stored at −80 °C until use.

**Negative staining TEM**. Purified sEVs were thawed and resuspended in 2% paraformaldehyde (PFA). Five μl of the samples were deposited on Formvar/carbon-coated grids for 20 min. The grids were washed 3 × 1 min with 1× PBS, then incubated with 1% glutaraldehyde for 5 min, followed by a washing step of 8 × 1 min with distilled water. The samples were stained in a drop of uranyl-oxalate solution (pH 7.0) for 5 min, then incubated with 4% uranyl acetate (pH 4.0) and 2% methylcellulose for 10 min on ice, protected from light. The excess of uranyl acetate and methylcellulose were removed by blotting on filter paper. The grids were dried for 5–10 min in air and examined by TEM, FEI Tecnai™ G2 (ThermoFisher Scientific, USA), operated at 80 kV.

**Nanoparticle tracking analysis (NTA)**. Nanoparticle tracking analysis was conducted using a Nano sight LM10HSB system equipped for fast video capture and particle tracking to determine the vesicle size distributions. Each sample was diluted 500-fold and analyzed in 5 runs each time for 30 s, recorded with a syringe speed of 50 using camera level 10, detection threshold 8, and the auto minimum

expected particle size and auto jump distance in analytical software NTA version 3.0 package.

**Isolation of sEVs' surface protein**. Volumes of 500 μl of SF-sEVs and PC3 sEVs, at a concentration of 2 mg/ml, were washed twice with PBS and ultracentrifuged at $112,000 \times g$, for 120 min. The pellet was resuspended in 1× PBS supplemented with protease inhibitors and ultracentrifuged at $112,000 \times g$ for an additional 120 min. The pellets were then resuspended and incubated in 500 μl 1× PBS (pH 8.0) supplemented with protease inhibitors and 1 mM EZ-Link sulfo-NHS-SS-Biotin (Pierce, Rockford, IL, USA) for 30 min on ice. Unreacted biotin was quenched by adding Tris-HCl to a final concentration of 50 mM, and incubated for 15 min. In order to remove free biotin, biotinylated SF-sEVs and PC3 sEVs were diluted in PBS and ultracentrifuged at $112,000 \times g$ for 120 min. The pellets were then resuspended in lysis buffer (6 M urea, proteases inhibitors, 1% n-octyl-β-D-glucopyranoside (β-OG) in PBS) and incubated for 1 h on ice. To improve protein solubilization, the samples were vortexed every 5 min for 5 s during the incubation time and then sonicated for 30 min. The lysates were centrifuged at $10,000 \times g$ at 4 °C for 10 min to remove debris. A fraction of the lysates were stored (total lysate), while the remainder was diluted 10-fold in 1× PBS supplemented with protease inhibitors, divided into three tubes (each tube contains 300 μg of total lysed sEVs) and incubated with 5 mg streptavidin magnetic beads (at the concentration of 10 mg/ml; Dynabeads MyOne™, Invitrogen) at room temperature (RT) with end-over-end mixing for 60 min. Streptavidin beads containing biotinylated surface proteins were then collected using a magnet and washed three times with 1× PBS containing protease inhibitors, while the supernatant containing the unbiotinylated proteins (total minus surface) was stored at −20 °C. Proteins were eluted from beads by 60 min incubation at RT in 50 mM DTT in PBS, with end-over-end mixing. Eluted proteins (surface) were then separated from the beads using a magnet and stored at −20 °C. Total protein concentrations for all the samples were determined by Dot-it-Spot-it (Maplestone, Knivsta, Sweden) according to the manufacturer's instructions. Before proceeding with HRMS, sample qualities were checked by gel electrophoresis. Samples were diluted in 4× LDS sample buffer (Invitrogen), loaded on a 4–12% Bis-Tris gel (Invitrogen, Carlsbad, CA, USA), and run at 200 V for 50 min. The gel was stained by silver staining (GE Healthcare, Piscataway, NJ, USA) following the manufacturer's instructions.

**Western blot**. SF-sEVs, PC3 sEVs, and PC3 cells were lysed in RIPA buffer (Santa Cruz Biotechnology, Santa Cruz, CA) supplemented with protease inhibitors, separated by SDS-PAGE under reducing conditions, and blotted with iBlot2 dry blotting system (ThermoFisher Scientific). LI-COR TBS blocking buffer (LI-COR Biosciences, Lincoln, NE, USA) was used for blocking and antibody incubation. The proteins were analyzed using 1.3 μg/ml anti-TSG101 antibody, 0.5 μg/ml anti-CD9 antibody, 0.5 μg/ml anti-CD81 antibody, 0.5 μg/ml anti-CD63 antibody and 0.1 μg/ml anti-calnexin antibody, which were detected using 50 ng/ml donkey anti-mouse IgG IRDye 680LT or 75 ng/ml donkey anti-rabbit IgG IRDye 800CW as secondary antibodies and analyzed using Odyssey scanner from LI-COR. All information for the antibodies are listed in Supplementary Table 1.

**Protein digestion and nanoLC MS/MS protein analysis**. Proteins in the solution were reduced, alkylated and on-filter digested by trypsin. Dried peptides were dissolved in 0.1% formic acid and diluted before injection to load an equal amount of peptides for each sample. Peptides were separated in reverse-phase on a nanoLC C18-column, applying a 90 min gradient and electrosprayed on-line to a HRMS QE-Orbitrap mass spectrometer (Thermo Finnigan). Tandem mass spectrometry was performed by applying higher-energy collisional dissociation (HCD). Data search was performed using the Sequest algorithm, embedded in Proteome Discoverer 1.4 (ThermoFisher Scientific) against a FASTA Uniprot database (Homo sapiens, reviewed, released May 2019, 20421 entries). The search parameters were set to Taxonomy: Homo sapiens; Enzyme: Trypsin. The fixed modification was Carbamidomethyl (C), while the variable modifications were Oxidation (M), and Deamidated (NQ). The search criteria for protein identification were set to at least two matching peptides and a 95% confidence level per protein.

**Detection of surface proteins by Exo-PLA**. Exo-PLA was performed by adapting the protocol published by Löf et al.[45]. Briefly, a mixture of capturing antibodies mouse monoclonal anti-CD9, CD63 and dipeptidyl peptidase-4 (CD26) (Supplementary Table 1), were immobilized via conjugated DNA oligonucleotides to oligonucleotides immobilized on magnetic beads as previously described. A list of oligonucleotides are provided in Supplementary Table 2[46]. A mixture of the three antibodies was used in order to maximize the efficiency of sEV capture. A number of PLA probes—oligonucleotide-conjugated antibodies—were used to analyze the surface proteins composition of captured sEVs. Exo-PLA gives rise to a signal when pairs of PLA probes in close proximity generate DNA circles that template the formation of RCA products, which can be visualized with fluorophore-labeled oligonucleotides. Here, one PLA probe represented a mixture of antibodies directed against two well-known exosomal surface markers, neprilysin (CD10) and aminopeptidase N (CD13), by coupling the same DNA oligonucleotide to the two antibodies. A second PLA probe was prepared by coupling the counterpart DNA oligonucleotide separately to antibodies against the following sEV-specific targets

to identify subpopulations of sEVs: ACPP, PSA, PSMA, PTGDS, AKAP4, SEMG1, GAPDS, CRISP1 and CD59 (Supplementary Data 1). The sEVs labeled with the different RCA products were analyzed by flow cytometry on BD FACS Aria III or BD LSR Fortessa instruments (BD biosciences). Gates for positive signals in different populations were set using negative controls containing all the experimental reagents except the targets (SF-sEVs or PC3 sEVs). The gating strategy is explained in Supplementary Fig. 6a. For each reaction, 1 µl RCA products-labeled sEVs were sampled for control by fluorescence microscopy. Images were acquired with a 40× Plan-Apochromat objective, NA 1.3, on a Zeiss Axio Imager Z2 microscope and digital camera Hamamatsu C11440.

**Detection of surface proteins by SP-PLA**. SP-PLA was performed as previously described[49,81] with some modifications. Briefly, 25–35 µg of antibody raised against the following proteins was coupled to 5 mg of Dynabeads M-270 Epoxy magnetic beads according to the manufacturer's instructions (Cat. No. 14301, ThermoFisher Scientific): Calnexin, TSG101, PTGDS, AKAP4, CD63, SEMG1, and CRISP1 (Supplementary Table 1). The two SP-PLA probes for detection of the target sEVs were constructed by coupling the streptavidin-conjugated oligonucleotide SLC1 and SLC2 (Supplementary Table 2) to biotinylated antibodies against CD26 and CD9, respectively, at a 1:1 ratio. SF-sEVs and PC3 sEVs were diluted in the assay buffer (1 mM D-biotin, 0.1% BSA, 0.05% Tween-20, 100 nM goat IgG, 100 µg/ml salmon sperm DNA, 5 mM EDTA in 1x PBS) in a 10-fold serial dilution, 100 µg/ml–100 pg/ml for SF-sEVs, and 70 µg/ml–70 pg/ml for PC3 sEVs. A 10-fold dilution series of 20 µg/ml–20 pg/ml was applied when targeting CD63 in SF-sEVs. All samples were analyzed in triplicates, captured on 200 ng/µl of antibody-coupled beads, and labeled with 500 pM of each SP-PLA probe in a reaction volume of 50 µl assay buffer. Real-time PCR was performed in 25 µl volumes of amplification buffer (1× PCR buffer, 2.5 mM MgCl$_2$, 0.25× Sybr Green, 0.1 µM BioFwd primer, 0.1 µM BioRev primer, 0.1 µM BioSplint, 0.08 mM ATP, 0.2 mM dNTPs (with dUTP), 0.03 U/µl Platinum Taq DNA polymerase, 0.01 U/µl T4 ligase and 0.002 U/µl Uracil-DNA glycosylase) on a QuantStudio 6 Real-Time PCR system (Applied Biosystems) programmed at 95 °C for 10 min, followed by 45 cycles of 95 °C for 15 s and 60 °C for 1 min.

**Statistics and reproducibility**. In this study, a total of two anonymized samples of SF-sEVs were analyzed, each was prepared by pooling seminal plasma samples from five individuals. Peptide spectrum match (PSM) values were used for qualitative and semi-quantitative analysis. However, values were transformed by normalizing to the total nPSM. Missing values were treated as missing not at random (MNAR)[82] and substituted with the value of 0.05 (single-value imputation approach). The degree of enrichment for surface proteins was calculated as the ratio of nPSM between the surface and total minus surface fractions. An average value was calculated when replicates were available. Replicates 1 and 2 of the total lysates of SF-sEVs (Supplementary Figs. 1a and 2a) represented both biological and technical replicates (Rep1, from sample 1 and Rep2, from sample 2), while the three replicates for surface and total minus surface included one biological replicate and two technical replicates: Rep 1, SF-sEVs purified from sample 1: Rep 2 and Rep 3: SF-sEVs purified from sample 2. The significance of the calculated ratio was assessed by t-test. The variability across replicates was calculated as [(number of proteins uniquely identified for each sample/ total number of proteins) × 100]. Data analysis and representation were carried out using the R environment for statistical computing and visualization and the software GraphPad Prism (Graph Pad Software Inc.). Proteins identified were compared against the Exocarta database. The file for protein annotation listed in the Supplementary Data 1 was downloaded from Uniprot (http://www.uniprot.org/uploadlists/). The database for annotation, visualization, and integrated discovery Bioinformatics Resources 6.8, NIAID/NIH, and the PANTHER classification system[83] were used for Gene Ontology (GO) annotation of the proteins enriched in the different fractions (ratio ≥2, number of replicates ≥2). The Exo-PLA data were analyzed with BD FACS Diva software 8.0 (BD biosciences). For SP-PLA, all samples were analyzed in triplicates and data were analyzed with Microsoft Excel software. Figure 1a was generated manually using Inkscape software.

**Reporting summary**. Further information on research design is available in the Nature Portfolio Reporting Summary linked to this article.

## Data availability
The mass spectrometry proteomics data have been deposited to the ProteomeXchange Consortium via the PRIDE[84] partner repository with the dataset identifier PXD037791. The source data underlying Figs. 3, 4 and Supplementary Figs. 1, 2, 4, 5 are provided in Supplementary Data 1. The source data underlying Supplementary Fig. 3 is provided in Supplementary Data 2. Uncropped and unedited western blot pictures are available as Supplementary Fig. 7.

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

## Acknowledgements

We are grateful to the Mass Spectrometry-based Proteomics and BioVis facilities at Uppsala University for their assistance and Dr. Ahmed Ibrahim for help with the NTA. This work was supported by the SciLifeLab, Swedish Research Council under Grants 2020-02258, 2017-04152, 2018-02943, 2018-06156, 2018-02806, and 2015-4870, Torsten Söderbergs Stiftelse under Grant M130/16, Swedish Prostate Cancer Federation, Exodiab, The Swedish Cancer Foundation, The Cancer Research Foundations of Radiumhemmet and the Swedish Foundation for Strategic Research under Grant SB16-0046. The funders had no role in any part of this study.

## Author contributions

M.K.M., E.M.D., U.L., C.F., and J.B. conceived the idea and planned the study. E.M.D., A.A., Q.S., A.L., O.L., and L.D. participated in EVs production and purification. E.M.D. and C.F. carried out protein enrichment experiments. R.G. planned and carried out western blot and SP-PLA experiments and data analysis. J.B. was responsible for the MS analysis. C.F. planned and carried out the data analysis of MS data. E.M.D. and S.G. aided with NTA. E.M.D., R.I., and L.L. carried out electron microscopy and Exo-PLA experiments and data analysis with the help of A.A. C.F., E.M.D., R.G., and L.L. wrote the paper with the supervision of M.K.M. All authors contributed to the final version of the manuscript.

## Funding

## Competing interests

Ulf Landegren is a shareholder of Navinci and Olink Proteomics, having rights to the PLA technology. Other authors declare no competing interests.
