## [Peer Review File · Communications Biology]

Reviewers' comments:

Reviewer #1 (Remarks to the Author):

This study develops a strategy that combines high resolution mass spec and proximity ligation assay, to identify surface proteins of extracellular vesicles.

Major points:

The approach is interesting and innovative, but its significance is unclear without orthogonal approaches and additional information on the technical aspects of the method.

The methods used to isolate EVs are pretty standard and among the best available to separate EVs from soluble proteins, but the characterization of these EVs is completely missing from the main figures.

What is the functional significance of the newly identified surface proteins?

The use of the term exosome is discouraged when it is impossible to know if all the vesicles are derived from multivesicular bodies or rather are of the type of the ectosomes. The term EVs is more appropriate.

One major concern is the almost complete lack of validation. Also, it is not clear how easy this approach can be implemented in all laboratories. Additionally, without any functional study it is difficult to understand if this approach will be useful to identify new therapeutic targets.

Minor points:

How can "total minus surface" have more proteins than "total lysate"?

The combination of methods used to isolate EVs should separate them from proteins. However, Fibronectin has been identified as a protein attached to the surface but external to the EVs. It would be interesting to know and show if fibronectin was also identified in the SEC fractions containing soluble proteins.

FASN is known as a large cytosolic enzyme, but it appears as surface protein. The dataset could be analyzed in comparison with publicly available datasets obtained with different methods.

Reviewer #2 (Remarks to the Author):

The manuscript describes a workflow to identify surface-, and cargo enriched proteins in prostasomes; the result is a compendium/encyclopedia of proteins. This info can be very useful for 'the field'. The methodological part is strong and particularly the fact that they use replicate samples provides more robust data. The biological interpretations are somewhat less convincing.

1. The study design; comparing "prostasomes" from seminal fluid with those from PC3 cells is a bit farfetched. PC3 cells are AR negative and bear very little similarity to human PrCa which is mostly strong AR positive.
2. The majority of surface enriched proteins are classical secreted proteins (eg seminogelin, KLK3/PSA), cytoskeletal proteins etc. The biological basis for this is not discussed, eg nonspecific adhesion to EVs, as is documented for nucleic acids on EVs/prostasomes.
3. Minor; the term prostasomes is elusive; at least 50 % of the seminal fluid is derived from seminal vesicles (producing eg seminogelin). Describe as what they are SFosomes

Reviewer #3 (Remarks to the Author):

In this study Doulabi and colleagues used surface biotinylation, proximity ligation assay combined with proteomics to identify surface markers of prostate-derived exosomes. They identified 74 surface markers and validate several of them.

This study constitutes a significant resource for the EV research community and may help to identify key proteins involved in EV biology, especially towards prostate/Sperm communications.

The methods and results are well described, control experiments are also properly performed to validate and support the authors conclusions.

The following point should be addressed/

-The authors mentioned line 141 of page 6 Supplementary Figure 1D which is not included in the actual figure.

-The authors used the term "exosome". Since the MVB origin is not demonstrated, they should consider using the generic term "EV" to avoid endless semantic debate that may affect the future citation and recognition of the study.

-Finally, several of the 74 identified surface markers are also protein found in the cytosol. Which is puzzling. This has been a recurrent issue in many other proteomics studies. The authors might consider to systematically report, based on published data or software prediction, for each of the 74 "surface" proteins if they are either i) transmembrane and PM or MVB localized, ii) know secreted proteins that follow the secretory pathway and that may later bind PM-localized receptor ii) mostly known as cytosolic proteins that may be released outside the cell via unconventional secretion associated or not with EV secretion. This could be included in the main figure and in the discussion and would certainly constitute a important point to guide the scientific community towards putative new candidates that may be involved in EV biological functions. This would certainly strengthen the impact of the study and add significant information on this resource study.

Dear Dr. Rogers,

Please find below our point-by-point responses to the reviewers' comments.

Your sincerely,

Masood Kamali-Moghaddam with coauthors

Reviewer #1 (Remarks to the Author):

This study develops a strategy that combines high-resolution mass spec and proximity ligation assay, to identify surface proteins of extracellular vesicles.

Major points:

- 1) The approach is interesting and innovative, but its significance is unclear without orthogonal approaches and additional information on the technical aspects of the method.

We believe our study indeed includes orthogonal approaches by design, as mass spectrometry (MS) and antibody-based assays are commonly used to confirm each other (1, 2). In the workflow we present here, we demonstrate the use MS to discover extracellular vesicle surface proteins (primary method). This was followed by experiments using the two affinity-based assays SP-PLA and Exo-PLA for validation. Using SP-PLA and Exo-PLA, we validated a selected number of targets with prostate-specific expression and with relevance for the study. The main scope of this study was to establish the workflow but not to deeply evaluate and validate all identified proteins. We have now added a sentence in the introduction [line 92] and additional references (1, 2) to further explain how our work follows an orthogonal strategy to confirm our findings.

- 2) The methods used to isolate EVs are pretty standard and among the best available to separate EVs from soluble proteins, but the characterization of these EVs is completely missing from the main figures.

The EVs used in this study are characterized in detail in accordance with MISEV 2018. We have now moved this information from supplementary data to the main text in the legend of the new Figure 2 with data on western blot, NTA, TEM, and gel electrophoresis.

- 3) What is the functional significance of the newly identified surface proteins?

As mentioned above, the scope of this study was mainly to establish a workflow for characterization of EVs with regard to their surface protein. However, we have now added information about confirmed or putative functions of the 74 newly identified surface proteins as Supplementary Table 3. In addition, the functions and networks of these proteins were further analyzed using the STRING toolbox based on seven criteria: involvement in the immune system, male reproductive system or disease development, expression in cytosol, extracellular vesicles, multivesicular bodies and seminal vesicles. The results are presented as a new paragraph in the Results section (Lines 192-198) and in Supplementary Figure 3.

- 4) The use of the term exosome is discouraged when it is impossible to know if all the vesicles are derived from multivesicular bodies or rather are of the type of the ectosomes. The term EVs is more appropriate.

The entire manuscript has now been revised and the terms extracellular vesicles (EVs) and small-EVs (sEV, when appropriate) are used throughout.

- 5) One major concern is the almost complete lack of validation. Also, it is not clear how easy this approach can be implemented in all laboratories. Additionally, without any functional study it is difficult to understand if this approach will be useful to identify new therapeutic targets.

As mentioned above, we have provided validation data for some of the most relevant proteins identified by MS, using two antibody-based methods. SP-PLA and Exo-PLA. Our strategy thus involves two steps (i) a broad proteome profiling using MS; this step is applicable for research laboratory equipped with the required MS instrumentation themselves or through a service provider, allowing broad identification of proteins; followed by (ii) antibody-based detection of EVs using specific surface protein targets by a PLA-based immunoassay, which are proven to be easily implemented in any pre-clinical and clinical laboratory upon appropriate analytical validation.

Minor points:

- 6) How can “total minus surface” have more proteins than “total lysate”?

Thank you for bringing up this important observation. As shown in Supplementary Figure 2, 1414 proteins were identified in total proteasome lysate and 1460 in total minus surface.

Despite the fact that less proteins would be present in the total minus surface fraction, it is nonetheless surprising that the two samples show similar numbers of proteins, considering the common variability between MS runs. We note that in the three MS experiments for “total minus surface” samples 1132, 1208, and 1243 proteins were recorded. Between the three replicates a total of 1460 proteins were identified since not all proteins were identified in all three replicates.

The two samples (total proteasome lysate and total minus surface) are similar in number of proteins for more than one reason:

Prostasome purification of biotinylated surface proteins using beads cannot quantitatively capture surface proteins. As we explained in the Discussion, purification using streptavidin-coated beads probably only captures a fraction of proteins present in the lysate, accounting for the variable identification of proteins between experiments. To achieve a more complete depletion of biotinylated proteins from the total lysate a greater number of beads could be used. However, the procedure would be costlier and potentially also limited by the total number of peptides that can be recorded in any MS run.

As described in the Materials and methods section (Lines 457-462) “Protein digestion and nanoLC MS/MS analysis”), an equal amount of peptides was injected for each sample run, preventing direct comparisons between numbers of proteins identified in the different fractions, the two samples were brought again at the same concentration. Here, some highly abundant proteins might be removed in favor of detection of some low abundant proteins.

- 7) The combination of methods used to isolate EVs should separate them from proteins. However, Fibronectin has been identified as a protein attached to the surface but external to the EVs. It would be interesting to know and show if fibronectin was also identified in the SEC fractions containing soluble proteins.
- 8) FASN is known as a large cytosolic enzyme, but it appears as surface protein. The dataset could be analyzed in comparison with publicly available datasets obtained with different methods.

We believe the two questions 7 and 8 are related to what is referred to as the protein corona. We have now added a paragraph in the Discussion describing the possibility that some of the identified surface proteins may, indeed, be part of this protein corona, which can be attached to the EV surface as a secondary layer of proteins (Lines 352-355). The SEC wash fractions are too dilute to fully address the reviewer's question but we hope that future works will resolve the issue.

In particular, FASN has been previously reported to be present on various cells including prostate cell lines and on different EVs (3, 4).

Reviewer #2 (Remarks to the Author):

The manuscript describes a workflow to identify surface-, and cargo enriched proteins in prostasomes; the result is a compendium/encyclopedia of proteins. This info can be very useful for 'the field'. The methodological part is strong and particularly the fact that they use replicate samples provides more robust data. The biological interpretations are somewhat less convincing.

- 1) The study design; comparing "prostasomes" from seminal fluid with those from PC3 cells is a bit farfetched. PC3 cells are AR negative and bear very little similarity to human PrCa which is mostly strong AR positive.

We have now removed the direct comparison between prostasomes and PC3 sEVs, including the previous Supplementary Figure 6.

- 2) The majority of surface enriched proteins are classical secreted proteins (eg seminogelin, KLK3/PSA), cytoskeletal proteins etc. The biological basis for this is not discussed, eg nonspecific adhesion to EVs, as is documented for nucleic acids on EVs/prostasomes.

We are thankful to the reviewer for the opportunity to clarify this point. Gene Ontology (GO) terms, describing protein localization, involvement in biological processes and pathways, molecular function and tissue specificity for all the proteins identified in this study, were previously fully listed in Supplementary Table 1. We have now added a column in Supplementary Table 2 to underscore that actually 43 out of 74 proteins are membrane proteins according to GO annotation (Lines 189-191). As seen in Supplementary Table 1, proteins may be associated to more than one GO terms due to multiple function. Further, we now discuss protein networks and interactions investigated using STRING, and the proteins' potential roles in the immune system, male reproductive system and disease biology, cytosol, seminal vesicles, multivesicular body, extracellular vesicles (Supplementary Figure 3 and Supplementary Table 3). In addition, another new paragraph describes the possibility that proteins may adhere to the EVs perhaps as a layer of proteins absorbed from body fluids, commonly referred to as s protein corona (Lines 352-355)(5).

- 3) Minor; the term prostasomes is elusive; at least 50 % of the seminal fluid is derived from seminal vesicles (producing eg seminogelin). Describe as what they are SFosomes

We realize there are different nomenclatures but to avoid further confusion, we propose to continue using the term prostasomes to follow the terminology used since the late 60's as a collective term for EVs isolated from seminal fluids, in particular when we hope to use these EVs as biomarkers for prostate cancer.

Reviewer #3 (Remarks to the Author):

In this study Doulabi and colleagues used surface biotinylation, proximity ligation assay combined with proteomics to identify surface markers of prostate-derived exosomes. They identified 74 surface markers and validate several of them.

This study constitutes a significant resource for the EV research community and may help to identify key proteins involved in EV biology, especially towards prostate/Sperm communications.

The methods and results are well described, control experiments are also properly performed to validate and support the author's conclusions.

We appreciate the positive comments from the reviewer and the acknowledgment to our commitment to use appropriate controls and replicates in order to generate good quality data.

The following point should be addressed/

- 1) The authors mentioned line 141 of page 6 Supplementary Figure 1D which is not included in the actual figure.

Thanks for pointing out this error. All the figures have now been rearranged and numbered correctly.

- 2) The authors used the term "exosome". Since the MVB origin is not demonstrated, they should consider using the generic term "EV" to avoid endless semantic debate that may affect the future citation and recognition of the study.

As recommended, we have now revised the manuscript and the term "exosome" has been replaced by either EV or as appropriate small-EVs.

- 3) Finally, several of the 74 identified surface markers are also protein found in the cytosol. Which is puzzling. This has been a recurrent issue in many other proteomics studies. The authors might consider to systematically report, based on published data or software prediction, for each of the 74 "surface" proteins if they are either i) transmembrane and PM or MVB localized, ii) know secreted proteins that follow the secretory pathway and that may later bind PM-localized receptor ii) mostly known as cytosolic proteins that may be released outside the cell via unconventional secretion associated or not with EV secretion. This could be included in the main figure and in the discussion and would certainly constitute a important point to guide the scientific community towards putative new candidates that may be involved in EV biological functions. This would certainly strengthen the impact of the study and add significant information on this resource study.

As recommended, we now discuss the presence and putative functions of the 74 identified surface proteins in greater detail as a new paragraph in the Results section (Lines 189-191), in Supplementary

Table 3 and Supplementary Figure 3. We discuss the possible presence of some proteins as a second layer on the surface, a phenomenon known as “corona proteins”(5).

References

1. Zhou Q, Andersson R, Hu D, Bauden M, Kristl T, Sasor A, et al. Quantitative proteomics identifies brain acid soluble protein 1 (BASP1) as a prognostic biomarker candidate in pancreatic cancer tissue. *EBioMedicine*. 2019;43:282-94.
2. Andersson A, Remnestål J, Nellgård B, Vunk H, Kotol D, Edfors F, et al. Development of parallel reaction monitoring assays for cerebrospinal fluid proteins associated with Alzheimer's disease. *Clinica Chimica Acta*. 2019;494:79-93.
3. Hosseini-Beheshti E, Pham S, Adomat H, Li N, Guns EST. Exosomes as biomarker enriched microvesicles: characterization of exosomal proteins derived from a panel of prostate cell lines with distinct AR phenotypes. *Molecular & Cellular Proteomics*. 2012;11(10):863-85.
4. Lazar I, Clement E, Ducoux-Petit M, Denat L, Soldan V, Dauvillier S, et al. Proteome characterization of melanoma exosomes reveals a specific signature for metastatic cell lines. *Pigment cell & melanoma research*. 2015;28(4):464-75.
5. Tóth EÁ, Turiák L, Visnovitz T, Cserép C, Mázló A, Sódar BW, et al. Formation of a protein corona on the surface of extracellular vesicles in blood plasma. *Journal of extracellular vesicles*. 2021;10(11):e12140.

Reviewers' comments:

Reviewer #1 (Remarks to the Author):

The authors have addressed several of the previously raised concerns. However some points still need attention.

The terminology has been changed from exosomes to EVs, which is highly recommended, however the characterization of these vesicles is still poor and the western blot display is suboptimal.

The term prostasome has been almost completely abandoned in the literature, and in fact papers cited by the authors are from several years ago. I would mention the term since these vesicles are derived from prostatic fluid and have been named prostasomes in the past, but then indicate that EVs will be used for all EVs.

Validation by flow cytometry for surface proteins and/or immune-capture would strengthen the study.

Reviewer #4 (Remarks to the Author):

The authors have convincingly addressed my previous concerns and comments.

Dear Dr. Rogers,

Please find below our point-by-point responses to the reviewers' comments.

Your sincerely,

Masood Kamali-Moghaddam with coauthors

Reviewer #1 (Remarks to the Author):

This study develops a strategy that combines high-resolution mass spec and proximity ligation assay, to identify surface proteins of extracellular vesicles.

Major points:

- 1) The approach is interesting and innovative, but its significance is unclear without orthogonal approaches and additional information on the technical aspects of the method.

We believe our study indeed includes orthogonal approaches by design, as mass spectrometry (MS) and antibody-based assays are commonly used to confirm each other (1, 2). In the workflow we present here, we demonstrate the use MS to discover extracellular vesicle surface proteins (primary method). This was followed by experiments using the two affinity-based assays SP-PLA and Exo-PLA for validation. Using SP-PLA and Exo-PLA, we validated a selected number of targets with prostate-specific expression and with relevance for the study. The main scope of this study was to establish the workflow but not to deeply evaluate and validate all identified proteins. We have now added a sentence in the introduction [line 92] and additional references (1, 2) to further explain how our work follows an orthogonal strategy to confirm our findings.

- 2) The methods used to isolate EVs are pretty standard and among the best available to separate EVs from soluble proteins, but the characterization of these EVs is completely missing from the main figures.

The EVs used in this study are characterized in detail in accordance with MISEV 2018. We have now moved this information from supplementary data to the main text in the legend of the new Figure 2 with data on western blot, NTA, TEM, and gel electrophoresis.

- 3) What is the functional significance of the newly identified surface proteins?

As mentioned above, the scope of this study was mainly to establish a workflow for characterization of EVs with regard to their surface protein. However, we have now added information about confirmed or putative functions of the 74 newly identified surface proteins as Supplementary Table 3. In addition, the functions and networks of these proteins were further analyzed using the STRING toolbox based on seven criteria: involvement in the immune system, male reproductive system or disease development, expression in cytosol, extracellular vesicles, multivesicular bodies and seminal vesicles. The results are presented as a new paragraph in the Results section (Lines 192-198) and in Supplementary Figure 3.

- 4) The use of the term exosome is discouraged when it is impossible to know if all the vesicles are derived from multivesicular bodies or rather are of the type of the ectosomes. The term EVs is more appropriate.

The entire manuscript has now been revised and the terms extracellular vesicles (EVs) and small-EVs (sEV, when appropriate) are used throughout.

- 5) One major concern is the almost complete lack of validation. Also, it is not clear how easy this approach can be implemented in all laboratories. Additionally, without any functional study it is difficult to understand if this approach will be useful to identify new therapeutic targets.

As mentioned above, we have provided validation data for some of the most relevant proteins identified by MS, using two antibody-based methods. SP-PLA and Exo-PLA. Our strategy thus involves two steps (i) a broad proteome profiling using MS; this step is applicable for research laboratory equipped with the required MS instrumentation themselves or through a service provider, allowing broad identification of proteins; followed by (ii) antibody-based detection of EVs using specific surface protein targets by a PLA-based immunoassay, which are proven to be easily implemented in any pre-clinical and clinical laboratory upon appropriate analytical validation.

Minor points:

- 6) How can “total minus surface” have more proteins than “total lysate”?

Thank you for bringing up this important observation. As shown in Supplementary Figure 2, 1414 proteins were identified in total proteasome lysate and 1460 in total minus surface.

Despite the fact that less proteins would be present in the total minus surface fraction, it is nonetheless surprising that the two samples show similar numbers of proteins, considering the common variability between MS runs. We note that in the three MS experiments for “total minus surface” samples 1132, 1208, and 1243 proteins were recorded. Between the three replicates a total of 1460 proteins were identified since not all proteins were identified in all three replicates.

The two samples (total proteasome lysate and total minus surface) are similar in number of proteins for more than one reason:

Prostasome purification of biotinylated surface proteins using beads cannot quantitatively capture surface proteins. As we explained in the Discussion, purification using streptavidin-coated beads probably only captures a fraction of proteins present in the lysate, accounting for the variable identification of proteins between experiments. To achieve a more complete depletion of biotinylated proteins from the total lysate a greater number of beads could be used. However, the procedure would be costlier and potentially also limited by the total number of peptides that can be recorded in any MS run.

As described in the Materials and methods section (Lines 457-462) “Protein digestion and nanoLC MS/MS analysis”), an equal amount of peptides was injected for each sample run, preventing direct comparisons between numbers of proteins identified in the different fractions, the two samples were brought again at the same concentration. Here, some highly abundant proteins might be removed in favor of detection of some low abundant proteins.

- 7) The combination of methods used to isolate EVs should separate them from proteins. However, Fibronectin has been identified as a protein attached to the surface but external to the EVs. It would be interesting to know and show if fibronectin was also identified in the SEC fractions containing soluble proteins.
- 8) FASN is known as a large cytosolic enzyme, but it appears as surface protein. The dataset could be analyzed in comparison with publicly available datasets obtained with different methods.

We believe the two questions 7 and 8 are related to what is referred to as the protein corona. We have now added a paragraph in the Discussion describing the possibility that some of the identified surface proteins may, indeed, be part of this protein corona, which can be attached to the EV surface as a secondary layer of proteins (Lines 352-355). The SEC wash fractions are too dilute to fully address the reviewer's question but we hope that future works will resolve the issue.

In particular, FASN has been previously reported to be present on various cells including prostate cell lines and on different EVs (3, 4).

Reviewer #2 (Remarks to the Author):

The manuscript describes a workflow to identify surface-, and cargo enriched proteins in prostasomes; the result is a compendium/encyclopedia of proteins. This info can be very useful for 'the field'. The methodological part is strong and particularly the fact that they use replicate samples provides more robust data. The biological interpretations are somewhat less convincing.

- 1) The study design; comparing "prostasomes" from seminal fluid with those from PC3 cells is a bit farfetched. PC3 cells are AR negative and bear very little similarity to human PrCa which is mostly strong AR positive.

We have now removed the direct comparison between prostasomes and PC3 sEVs, including the previous Supplementary Figure 6.

- 2) The majority of surface enriched proteins are classical secreted proteins (eg seminogelin, KLK3/PSA), cytoskeletal proteins etc. The biological basis for this is not discussed, eg nonspecific adhesion to EVs, as is documented for nucleic acids on EVs/prostasomes.

We are thankful to the reviewer for the opportunity to clarify this point. Gene Ontology (GO) terms, describing protein localization, involvement in biological processes and pathways, molecular function and tissue specificity for all the proteins identified in this study, were previously fully listed in Supplementary Table 1. We have now added a column in Supplementary Table 2 to underscore that actually 43 out of 74 proteins are membrane proteins according to GO annotation (Lines 189-191). As seen in Supplementary Table 1, proteins may be associated to more than one GO terms due to multiple function. Further, we now discuss protein networks and interactions investigated using STRING, and the proteins' potential roles in the immune system, male reproductive system and disease biology, cytosol, seminal vesicles, multivesicular body, extracellular vesicles (Supplementary Figure 3 and Supplementary Table 3). In addition, another new paragraph describes the possibility that proteins may adhere to the EVs perhaps as a layer of proteins absorbed from body fluids, commonly referred to as s protein corona (Lines 352-355)(5).

- 3) Minor; the term prostasomes is elusive; at least 50 % of the seminal fluid is derived from seminal vesicles (producing eg seminogelin). Describe as what they are SFosomes

We realize there are different nomenclatures but to avoid further confusion, we propose to continue using the term prostasomes to follow the terminology used since the late 60's as a collective term for EVs isolated from seminal fluids, in particular when we hope to use these EVs as biomarkers for prostate cancer.

Reviewer #3 (Remarks to the Author):

In this study Doulabi and colleagues used surface biotinylation, proximity ligation assay combined with proteomics to identify surface markers of prostate-derived exosomes. They identified 74 surface markers and validate several of them.

This study constitutes a significant resource for the EV research community and may help to identify key proteins involved in EV biology, especially towards prostate/Sperm communications.

The methods and results are well described, control experiments are also properly performed to validate and support the author's conclusions.

We appreciate the positive comments from the reviewer and the acknowledgment to our commitment to use appropriate controls and replicates in order to generate good quality data.

The following point should be addressed/

- 1) The authors mentioned line 141 of page 6 Supplementary Figure 1D which is not included in the actual figure.

Thanks for pointing out this error. All the figures have now been rearranged and numbered correctly.

- 2) The authors used the term "exosome". Since the MVB origin is not demonstrated, they should consider using the generic term "EV" to avoid endless semantic debate that may affect the future citation and recognition of the study.

As recommended, we have now revised the manuscript and the term "exosome" has been replaced by either EV or as appropriate small-EVs.

- 3) Finally, several of the 74 identified surface markers are also protein found in the cytosol. Which is puzzling. This has been a recurrent issue in many other proteomics studies. The authors might consider to systematically report, based on published data or software prediction, for each of the 74 "surface" proteins if they are either i) transmembrane and PM or MVB localized, ii) know secreted proteins that follow the secretory pathway and that may later bind PM-localized receptor ii) mostly known as cytosolic proteins that may be released outside the cell via unconventional secretion associated or not with EV secretion. This could be included in the main figure and in the discussion and would certainly constitute a important point to guide the scientific community towards putative new candidates that may be involved in EV biological functions. This would certainly strengthen the impact of the study and add significant information on this resource study.

As recommended, we now discuss the presence and putative functions of the 74 identified surface proteins in greater detail as a new paragraph in the Results section (Lines 189-191), in Supplementary

Table 3 and Supplementary Figure 3. We discuss the possible presence of some proteins as a second layer on the surface, a phenomenon known as “corona proteins”(5).

References

1. Zhou Q, Andersson R, Hu D, Bauden M, Kristl T, Sasor A, et al. Quantitative proteomics identifies brain acid soluble protein 1 (BASP1) as a prognostic biomarker candidate in pancreatic cancer tissue. *EBioMedicine*. 2019;43:282-94.
2. Andersson A, Remnestrål J, Nellgård B, Vunk H, Kotol D, Edfors F, et al. Development of parallel reaction monitoring assays for cerebrospinal fluid proteins associated with Alzheimer's disease. *Clinica Chimica Acta*. 2019;494:79-93.
3. Hosseini-Beheshti E, Pham S, Adomat H, Li N, Guns EST. Exosomes as biomarker enriched microvesicles: characterization of exosomal proteins derived from a panel of prostate cell lines with distinct AR phenotypes. *Molecular & Cellular Proteomics*. 2012;11(10):863-85.
4. Lazar I, Clement E, Ducoux-Petit M, Denat L, Soldan V, Dauvillier S, et al. Proteome characterization of melanoma exosomes reveals a specific signature for metastatic cell lines. *Pigment cell & melanoma research*. 2015;28(4):464-75.
5. Tóth EÁ, Turiák L, Visnovitz T, Cserép C, Mázló A, Sódar BW, et al. Formation of a protein corona on the surface of extracellular vesicles in blood plasma. *Journal of extracellular vesicles*. 2021;10(11):e12140.

REVIEWERS' COMMENTS:

Reviewer #1 (Remarks to the Author):

The authors have addressed most of the issues.